# Increased burden of familial-associated early-onset cancer risk among minority Americans compared to non-Latino Whites

Qianxi Feng[1], Eric Nickels[1,2], Ivo S Muskens[1], Adam J de Smith[1], W James Gauderman[1], Amy C Yee[1], Charite Ricker[3], Thomas Mack[1], Andrew D Leavitt[4], Lucy A Godley[5], Joseph L Wiemels[1]*

[1]Department of Preventive Medicine, USC Keck School of Medicine, Los Angeles, United States; [2]Children's Hospital Los Angeles, Los Angeles, United States; [3]Norris Comprehensive Cancer Center, USC Keck School of Medicine, Los Angeles, United States; [4]Departments of Medicine and Laboratory Medicine, University of California, San Francisco, San Francisco, United States; [5]Departments of Medicine and Human Genetics, The University of Chicago, Chicago, United States

*For correspondence: wiemels@usc.edu

Competing interests: The authors declare that no competing interests exist.

## Abstract

**Background:** The role of race/ethnicity in genetic predisposition of early-onset cancers can be estimated by comparing family-based cancer concordance rates among ethnic groups.

**Methods:** We used linked California health registries to evaluate the relative cancer risks for first-degree relatives of patients diagnosed between ages 0 and 26, and the relative risks of developing distinct second primary malignancies (SPMs). From 1989 to 2015, we identified 29,631 cancer patients and 62,863 healthy family members. We calculated the standardized incident ratios (SIRs) of early-onset primary cancers diagnosed in proband siblings and mothers, as well as SPMs detected among early-onset patients. Analyses were stratified by self-identified race/ethnicity.

**Results:** Given probands with cancer, there were increased relative risks of any cancer for siblings and mothers (SIR = 3.32; 95% confidence interval [CI]: 2.85–3.85) and of SPMs (SIR = 7.27; 95% CI: 6.56–8.03). Given a proband with solid cancer, both Latinos (SIR = 4.98; 95% CI: 3.82–6.39) and non-Latino Blacks (SIR = 7.35; 95% CI: 3.36–13.95) exhibited significantly higher relative risk of any cancer in siblings and mothers when compared to non-Latino White subjects (SIR = 3.02; 95% CI: 2.12–4.16). For hematologic cancers, higher familial risk was evident for Asian/Pacific Islanders (SIR = 7.56; 95% CI: 3.26–14.90) compared to non-Latino whites (SIR = 2.69; 95% CI: 1.62–4.20).

**Conclusions:** The data support a need for increased attention to the genetics of early-onset cancer predisposition and environmental factors in race/ethnic minority families in the United States.

**Funding:** This work was supported by the V Foundation for funding this work (Grant FP067172).

## Introduction

Both genetic and environmental factors play a role in the causes of early-onset cancer. Several well-defined genetic syndromes contribute to early-onset cancer risk, along with a wider array of common alleles that influence risk marginally and detected at the population level. As an example of the former, Li–Fraumeni syndrome caused by mutations in the tumor suppressor gene *TP53* is associated with an increased risk of a spectrum of cancers diagnosed at early ages (*Saletta et al., 2015*). An example of low penetrance common genetic variations associated with cancer risk includes *IKZF1* and *ARID5B* genes in pediatric acute lymphoblastic leukemia (ALL) (*Moriyama et al., 2015*). Both of

these classes of variants may vary in frequency by race/ethnic group and cluster by families (*Caswell-Jin et al., 2018*; *Walsh et al., 2013*; *Ricker et al., 2016*). Examination of cancer predisposition requires investigation in ethnic strata particularly where cancer incidence rates are known to differ as they do for many pediatric cancer types, such as leukemia (*Feng and de Smith, 2020*) and brain cancer (*Ostrom et al., 2019*).

In addition to primary cancers, incidence patterns of second independent malignancies may also provide a perspective of underlying genetic predisposition. Among childhood cancer survivors, more second primary malignancy (SPM) cases are observed among non-Latino Whites (NLW) than Latino subjects (*Brown et al., 2019*). This is also reflected in adult cancers, where Latino breast cancer survivors had lower risk of second cancers than NLW and NL Black women (*Calip et al., 2015*).

Germline pathogenic/likely pathogenic variants in cancer predisposition genes are found in approximately 10% of pediatric cancer patients (*Saletta et al., 2015*; *Zhang et al., 2015*; *Plon and Lupo, 2019*; *Ripperger et al., 2017*), and may be inherited or arise de novo. Highly penetrant inherited variants will contribute to clustering of cancer cases within the family. Shared environments within the family unit may also be considered alongside genetic risk as potential causes for family-based cancer concordance.

Familial concordance of a wide variety of cancers has been assessed using the Swedish Family-Cancer Database, leading to a deep understanding of familial relative risks (*Sud et al., 2019*; *Kharazmi et al., 2012*; *Hemminki and Czene, 2002*). The Victorian Paediatric Cancer Family Study in Australia also explored the cancer risks for relatives of children with cancer in a small population (*Heath et al., 2014*). In the United States, the Utah Population database may be the best-known population for studying familial risk (*Curtin et al., 2013*; *Kohli et al., 2019*; *Samadder et al., 2014*; *Wang et al., 2010*; *Goldgar et al., 1994*). Importantly, these studies largely comprise families of European ancestry, and therefore have not examined potential ethnic-specific familial risks. Here, we utilized linked population registries with over 64,000 individuals to quantify the familial risks (siblings and mothers) and the risks of early-onset SPMs in the highly diverse and large population of California. Risk patterns suggest that race/ethnic minority subjects in the United States may harbor a higher burden of familial risks for some types of cancers compared with the majority population – NLW.

## Materials and methods

### Source of data

We used linked population-based registries in California to evaluate the relative risks of early-onset cancers (0–26 years age of onset) for siblings and mothers of children, adolescents, and young adults (AYA) aged 0–26 diagnosed with cancer, as well as the relative risks of early-onset SPMs among the proband patients. The dataset was created by linking information from the California Cancer Registry (CCR) and California Birth Statistical Master File, allowing the capture of siblings and parents of cancer probands, along with their cancer incidence (*California Cancer Registry, 2018*). The linked dataset comprehensively encompassed all cancer cases 0–26 years old, as well as their sibling and mother cancers, diagnosed from 1989 to 2015 in California. Our upper age limit of 26 was set based on the available age range covered by this relatively young cohort. Overall, the dataset included a total of 121,571 individuals. The information on healthy siblings and mothers was available during the whole study period, whereas the information on fathers was not available until 2004 in the birth files and therefore is not included in the current analysis.

For the analysis of cancer familial risks, we included all primary incident cancer cases diagnosed from 1989 to 2015 among patients aged 0–26 years, with patient age-at-diagnosis limited by the study time period for which California maintained a statewide SEER gold-standard cancer registry. For the analysis of secondary cancer risks, we included all SPMs diagnosed over the same years and patient age ranges. Although the CCR only records primary malignancies, some misclassification of relapsed or recurrent disease is possible. A physician (EN) reviewed diagnosis codes of all the cases diagnosed after the first primary case to prevent the misclassification of relapsed first primary malignancies (FPMs) as SPMs. For both analyses, we classified the cancers into 12 broad groups and subgroups as defined by the International Classification of Childhood Cancer, Third edition (ICCC-3, November 2012) (https://seer.cancer.gov/iccc/iccc3.html). The abbreviations for cancer types are included in *Table 1*. We also grouped the cancers into hematologic or solid categories in the

**Table 1.** Abbreviations of the 12 broad groups defined by the International Classification of Childhood Cancer, Third edition.

| Abbreviation | Definition* |
| --- | --- |
| Leukemias | I. Leukemias, myeloproliferative diseases, and myelodysplastic diseases |
| Lymphomas | II. Lymphomas and reticuloendothelial neoplasms |
| CNS tumors | III. CNS and miscellaneous intracranial and intraspinal neoplasms |
| Neuroblastomas | IV. Neuroblastoma and other peripheral nervous cell tumors |
| Retinoblastoma | V. Retinoblastoma |
| Renal tumors | VI. Renal tumors |
| Hepatic tumors | VII. Hepatic tumors |
| Bone tumors | VIII. Malignant bone tumors |
| Sarcomas | IX. Soft tissue and other extraosseous sarcomas |
| GCT | X. Germ cell tumors, trophoblastic tumors, and neoplasms of gonads |
| Epithelial neoplasms | XI. Other malignant epithelial neoplasms and malignant melanomas |
| Other | XII. Other and unspecified malignant neoplasms |

*Cancers were classified into groups as defined by the International Classification of Childhood Cancer, Third edition (ICCC-3, November 2012) (https://seer.cancer.gov/iccc/iccc3.html).

analyses. Hematologic cancers were defined as leukemias and lymphomas. Solid cancers were defined as central nervous system (CNS) tumors, neuroblastomas, retinoblastomas, renal tumors, hepatic tumors, bone tumors, sarcomas, germ cell tumors, epithelial neoplasms, and other and unspecified malignant neoplasms.

## Statistical analysis

We quantified the relative risks for siblings and mothers, and the relative risks of SPMs by calculating the standardized incident ratios (SIRs) of a given cancer or of SPMs among the healthy siblings, and the SIRs of a given cancer among healthy mothers of probands using a previous published method (*Sud et al., 2019*). We defined a proband as a pediatric or AYA patient with a given cancer. Only one child/AYA in each family can be a proband, so that in families with two or more cases, the proband is defined as the patient with the earliest date of diagnosis. Given a proband with cancer, we calculated the SIRs for a sibling or a mother in the same family for all types of cancers. Separately, we calculated the SIRs for a sibling for the same type of cancer as the proband. We also stratified the analyses by self-identified race/ethnicity of the mother in each family. The SIRs in siblings, mothers or of SPMs can be denoted as:

$$SIR = \frac{O}{E} = \frac{\sum_{i=1}^{N} \sum_{j=1}^{n_i} D_{ij}}{\sum_{i=1}^{N} \sum_{j=1}^{n_i} \sum_{k=1}^{Kmax} \lambda_k t_{ijk}}$$

where N is the number of families, $n_i$ is the number of non-proband individuals of interest (siblings/SPMs/mothers) in family i, and Kmax is the total number of age intervals. The data for each individual includes a disease indicator ($D_{ij}$) and the number of years 'at risk' during the kth age interval ($t_{ijk}$). A given individual is defined to be at risk beginning at their age when the proband in their family is diagnosed and ending either when they become affected themselves or when they are censored due to end of study follow-up. For siblings and mothers, age was stratified into seven groups as 0, 0–4, 5–9, 10–14, 15–19, 20–24, and 25–29 years. For the calculation of SIRs within a given race group, $\lambda_k$ is the race-, sex-, and age-specific incidence rate of a given cancer. We compared the SIRs across race/ethnic groups with approximate chi-square tests. The approximate chi-square method compares the probability of occurrence of events in one group to another, based on a binomial distribution. This comparison is not related to the 95% confidence intervals (CIs) for the SIRs. We designated that all events occurred right at the middle point of each calendar year. We also stratified the analysis by 5 year age groups. The 95% CIs were calculated assuming a Poisson distribution for categories with less than 100 observed cases. For categories with more than 100 observed cases, we

adopted the method as indicated by *Breslow and Day, 1987* to calculate the 95% CIs, as suggested by Washington State Department of Health (*Guidelines for Using Confidence Intervals for Public Health Assessment, 2012*; *Breslow and Day, 1987*). Statistical analyses were performed using R software (v 3.6.0). Any two-sided p-value less than 0.05 was considered statistically significant. A supplement is included with this manuscript with more information on the statistical tests and computational codes used. Please access **Source Code File** for more information.

## Results

### Demographics of the study population

From 1989 to 2015, we identified a total of 29,249 pediatric and AYA patients with a primary malignancy, comprising 29,072 probands, 112 affected siblings (from 110 families), and 65 affected mothers. All siblings were diagnosed after the proband's diagnosis as defined, and 56 (86%) of the 65 mothers were diagnosed after the proband's diagnosis. We also identified 387 SPMs among all pediatric and AYA probands (*Table 2*).

### Familial relative risks of early-onset cancers

Overall, we found a 3.32-fold (95% CI: 2.85–3.85) increased relative risk of any cancer among siblings and mothers who have a proband with cancer in the same family. Briefly, we found a 2.97-fold (95% CI: 2.30–3.78) increased relative risk of any cancer given a proband with hematologic cancers and a 4.54-fold (95% CI: 3.82–5.35) increased relative risk of any cancer given a proband with solid cancers. When stratified by cancer type, higher relative risks among siblings and mothers were observed given probands with leukemias, lymphomas, CNS tumors, retinoblastomas, renal tumors, sarcomas, germ cell tumors (GCTs), epithelial neoplasms, and other unspecified neoplasms (*Figure 1A*).

For the relative risk of specific cancer types, we found a 2.68-fold (95% CI: 1.68–4.06) increased risk of hematologic cancers among siblings and mothers of a proband with hematologic cancer, and

**Table 2.** Selected demographic characteristic of probands, affected siblings, and second primary malignancies among the early-onset cancer patients in the linked population-based registries in California, 1989–2015.

| | Overall* | No. of probands | No. of affected siblings | No. of affected mothers | No. of second primaries[†] |
|---|---|---|---|---|---|
| Overall | 29,249 | 29,072 | 112 | 65 | 387 |
| Age at diagnosis (years)[‡] | | | No (%) | | |
| 0 | 2592 (8.75) | 2611 (8.98) | 10 (8.93) | 0 (0.00) | ≦5 (1.03) |
| 1–4 | 8683 (29.3) | 8719 (29.99) | 15 (13.39) | ≦5 (3.08) | 37 (9.56) |
| 5–9 | 5054 (17.06) | 5057 (17.39) | 12 (10.71) | 0 (0.00) | 67 (17.31) |
| 10–14 | 4224 (14.26) | 4180 (14.38) | 22 (19.64) | 0 (0.00) | 98 (25.32) |
| 15–19 | 4734 (15.98) | 4664 (16.04) | 37 (33.04) | 7 (10.77) | 90 (23.26) |
| 20+ | 4344 (14.66) | 3841 (13.21) | 16 (14.29) | 56 (86.15) | 91 (23.51) |
| Gender | | | | | |
| Male | 15,528 (52.40) | 15,467 (53.20) | 56 (50.00) | NA | 198 (51.16) |
| Female | 14,102 (47.59) | 13,605 (46.80) | 56 (50.00) | 65 (100.00) | 189 (48.84) |
| Race/ethnicity | | | | | |
| Latino (all races) | 13,281 (44.82) | 13,059 (44.92) | 51 (45.54) | 26 (40.00) | 159 (41.09) |
| Non-Latino White | 11,410 (38.51) | 11,193 (38.50) | 39 (34.82) | 17 (26.15) | 128 (33.07) |
| Non-Latino Black | 1772 (5.98) | 1716 (5.90) | 9 (8.04) | 7 (10.77) | 10 (2.58) |
| Non-Latino Asian/Pacific Islander | 2605 (8.79) | 2551 (8.77) | 12 (10.71) | 0 (0.00) | 47 (12.14) |
| Other | 563 (1.94) | 553 (1.90) | ≦5 (0.89) | 15 (4.62) | 43 (11.11) |

*All early-onset cancer patients diagnosed from 1989 to 2015 identified in the linked population-based registries in California.

[†]Number of second primary malignancies diagnosed from 1989 to 2015 within all children (probands and affected siblings, excluding mothers) with early-onset cancers in the linked population-based registries in California.

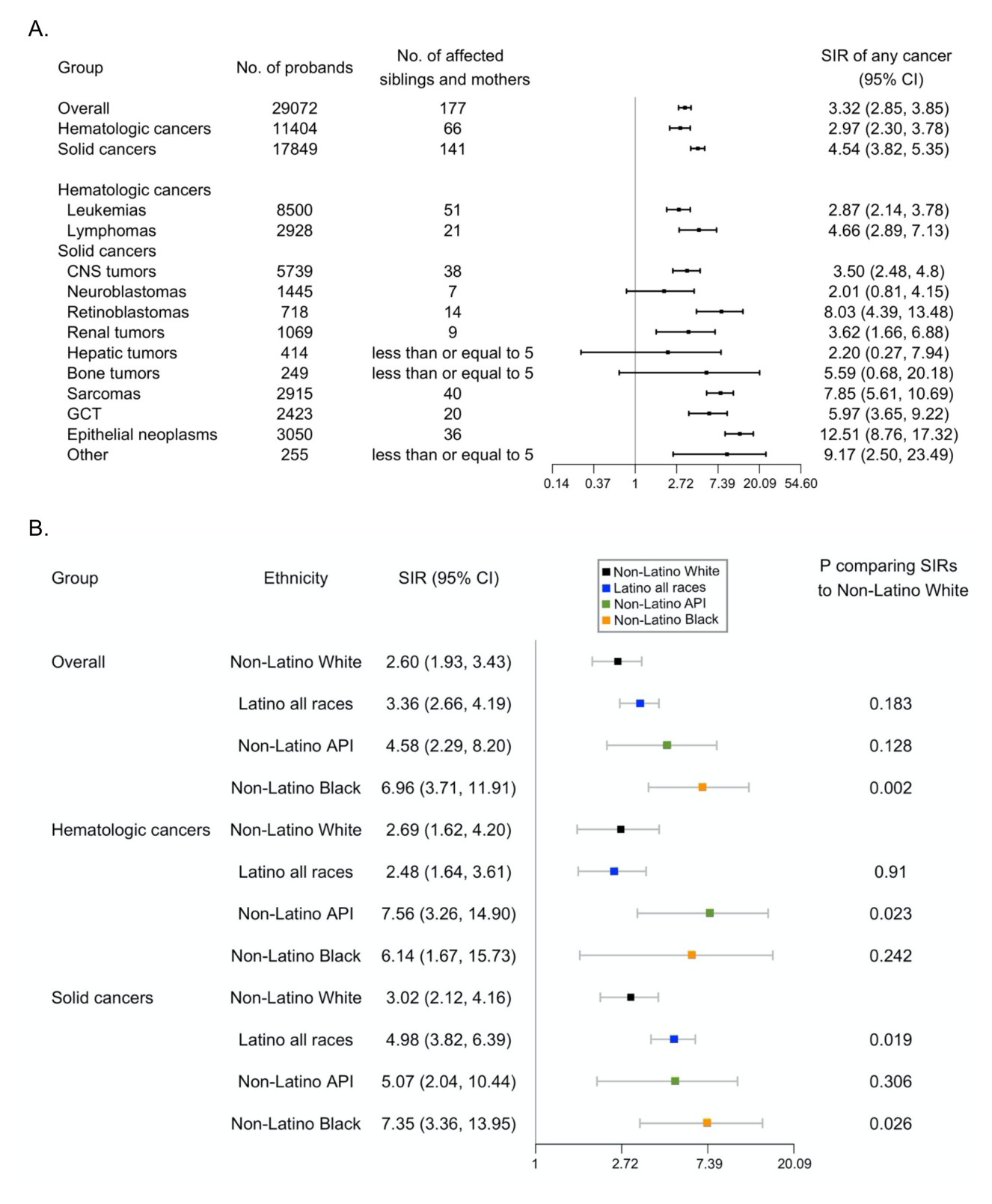

**Figure 1.** Relative risks of early-onset cancers among siblings and mothers. (**A**) Relative risks among siblings and mothers of any early-onset cancer (diagnosed under 26 years of age) given a proband with cancer, 1989–2015, California, USA. (**B**) Relative risks by ethnic group among siblings and mothers of any early-onset cancer (diagnosed under 26 years of age) given a proband with cancer, 1989–2015, California, USA. Cancers were classified into groups as defined by the International Classification of Childhood Cancer, Third edition (ICCC-3, November 2012) (https://seer.cancer.gov/iccc/

*Figure 1 continued on next page*

*Figure 1 continued*

iccc3.html). Hematologic cancers include leukemias and lymphomas. Solid cancers include CNS tumors, neuroblastomas, retinoblastomas, renal tumors, hepatic tumors, bone tumors, sarcomas, GCT, epithelial neoplasms, and others. The axis for SIR was natural log-transformed. SIR and 95% CI were not calculatable for cancers with zero observed case. p was calculated assuming a Poisson distribution. Abbreviations: SIR, standardized incidence ratio. CI, confidence interval. API, Asian/Pacific Islander.

a 6.78-fold (95% CI: 5.58–8.16) increased relative risk of solid cancers among siblings and mothers of a proband with solid cancer (*Supplementary file 1A*). Furthermore, leukemias, lymphomas, CNS tumors, retinoblastoma, sarcomas, GCT, and epithelial neoplasms exhibited statistically significantly increased relative risk for the same type of cancer as the proband (*Figure 2* and *Supplementary file 1B*).

When stratified by more finely defined cancer subtypes, increased relative risks of any cancer for siblings and mothers were observed given a proband with lymphoid leukemia, acute myeloid leukemia, Hodgkin lymphomas, non-Hodgkin lymphomas, astrocytomas, intracranial and intraspinal embryonal tumors, certain gliomas, certain specified intracranial and intraspinal neoplasms, nephroblastoma and other nonepithelial renal tumors, rhabdomyosarcomas, fibrosarcomas, peripheral nerve sheath tumors and other fibrous neoplasms, certain specified soft tissue sarcomas, malignant gonadal germ cell tumors, and certain unspecified carcinomas (*Supplementary file 1C*).

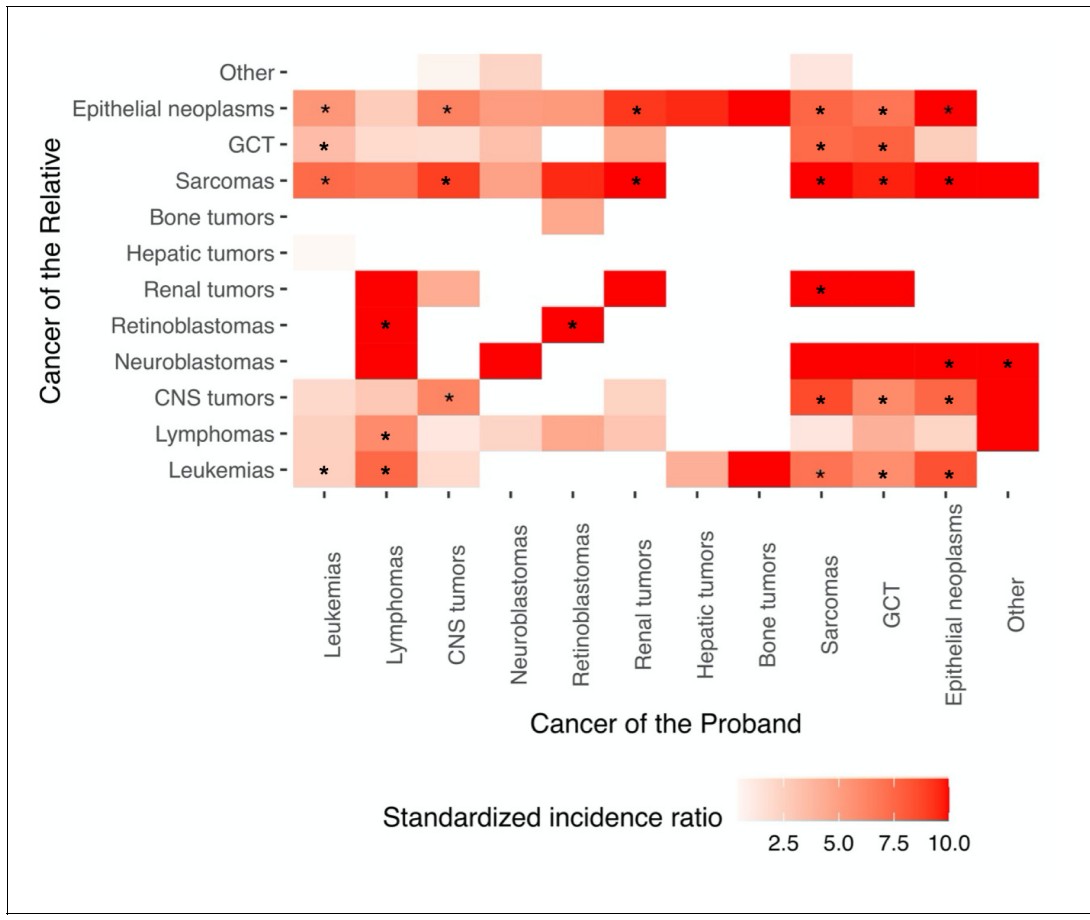

**Figure 2.** Relative risks of siblings and mothers of specific cancers. Cancers were classified into groups as defined by the International Classification of Childhood Cancer, Third edition (ICCC-3, November 2012) (https://seer.cancer.gov/iccc/iccc3.html). Standardized incidence ratios greater than 10 were recoded to 10. Siblings and mothers of a proband were diagnosed with cancer from 1989 to 2015 at 0–26 years of age. p was calculated assuming a Poisson distribution. Abbreviations: SIR, standardized incidence ratio. CI, confidence interval. GCT, germ cell tumors, trophoblastic tumors, and neoplasms of gonads.

When stratified by race/ethnicity, the relative risk of any cancer for siblings and mothers given a proband with solid cancer was significantly higher among Latino and non-Latino black subjects than NLW subjects (Latino: SIR = 4.98; 95% CI: 3.82–6.39; NLW: SIR = 3.02; 95% CI: 2.12–4.16; p=0.019) (*Figure 1B*). Non-Latino Asians/Pacific Islanders (API) had higher SIRs than NLW given a proband with hematologic cancer (SIR = 7.56; 95% CI: 3.26–14.90, p=0.023 compared to NLW), and non-Latino Blacks had higher SIRs than NLW given a proband with any cancer (SIR = 6.96; 95% CI: 3.71–11.91, p=0.002 compared to NLW) or solid cancer (SIR = 7.35; 95% CI: 3.36–13.95, p=0.026 compares to NLW) (*Figure 1B* and *Supplementary file 1D*). For the relative risk of the *same* category of cancers given a proband with that type of cancer, Latino subjects also showed higher relative risk of solid cancers than NLW subjects (Latino: SIR = 7.94; 95% CI: 5.89–10.47; NLW: SIR = 4.41; 95% CI: 2.99–6.25; p=0.012) (*Figure 1B* and *Supplementary file 1A*). Data on other minority groups (Asians, non-Latino Blacks) were too sparse to make this comparison.

## Relative risks of SPMs

Overall, SPMs among all childhood patients were enriched in families that exhibited familial risks; that is, those families with two or more primary cancer patients. We found 14 SPMs of 2432 members (0.58%) in the families that exhibited familial risks (two or more primary cancers) and 373 SPMs of 119,136 members (0.31%) in families that did not exhibit familial risk (p=0.023) (*Supplementary file 1E*).

For relative risks, we found a 7.27-fold increased risk of SPMs relative to the general population among children/AYAs with a FPM (SIR = 7.27; 95% CI: 6.56–8.03). Most primary cancer types were associated with an elevated relative risk of SPMs (*Figure 3A*). When stratified by race/ethnicity, a similar relative risk of all SPMs given a proband with cancer was observed among Latino subjects and NLW subjects (Latino: SIR = 6.85; 95% CI: 5.83–8.00; NLW: SIR = 6.65; 95% CI: 5.55–7.91; p=0.869) (*Figure 3B*). The relative risk of all SPMs given a proband with any cancer and solid cancers were higher among non-Latino API subjects compared to NLW subjects (any cancer, non-Latino API: SIR = 15.41; 95% CI: 10.85–21.24; p<0.001; hematologic cancers, non-Latino API: SIR = 18.90; 95% CI: 11.55–29.20; p<0.001) (*Figure 3B*).

For the relative risks of SPMs of the same cancer types as the FPM, we found elevated risks for both hematologic and solid cancers. When stratified by race/ethnicity, similar relative risks were observed among NLW subjects compared to Latino subjects given a proband with hematologic or solid cancer (*Supplementary file 1F*); numbers were too sparse to compare Asian and non-Latino Blacks.

## Discussion

To our knowledge, this is the first study to quantify the familial clustering risks and risks of SPMs among early-onset cancer patients with an emphasis on racial/ethnic differences. Using linked population registry data in the California population, we found that the risk for a sibling child/AYA or mother to have early-onset cancer was elevated once a proband was identified with an early-onset cancer. Likewise, the relative risks for SPMs were elevated among children/AYAs who contracted a first primary cancer. Due to the rarity of childhood cancers, the absolute risk of early-onset cancer is very small, but still higher among young siblings and mothers in the current study (0.074%) compared to general population (0.023%, calculated from SEER) of the same age group. The findings were consistent across race/ethnic groups; however, the magnitude was different. Latinos and non-Latino Blacks had higher sibling/maternal relative risks compared to NLWs for solid cancers, and APIs exhibited a higher risk of hematologic neoplasms.

Consistent with our results, a rich literature with a primary focus on European ancestry populations has reported excessive familial risks of hematologic malignancies (*Sud et al., 2019*), lymphomas (*Cerhan and Slager, 2015*; *Fallah et al., 2016*; *Madanat-Harjuoja et al., 2020*), brain tumors (*Couldwell and Cannon-Albright, 2017*; *Crump et al., 2015*), neuroblastomas (*Kamihara et al., 2017*), retinoblastomas (*Madanat-Harjuoja et al., 2020*; *Kamihara et al., 2017*), germ cell tumors (*Landero-Huerta et al., 2017*), sarcomas (*Lynch et al., 2003*), and melanomas (*Frank et al., 2017*). In terms of secondary cancers, studies have reported excessive risks of SPMs among of survivors of hereditary retinoblastoma (*Marees et al., 2008*), chronic myeloid leukemia (*Sasaki et al., 2019*), chronic lymphocytic leukemia (*Molica, 2005*), Hodgkin's lymphoma (*Baker, 2016*), non-Hodgkin's

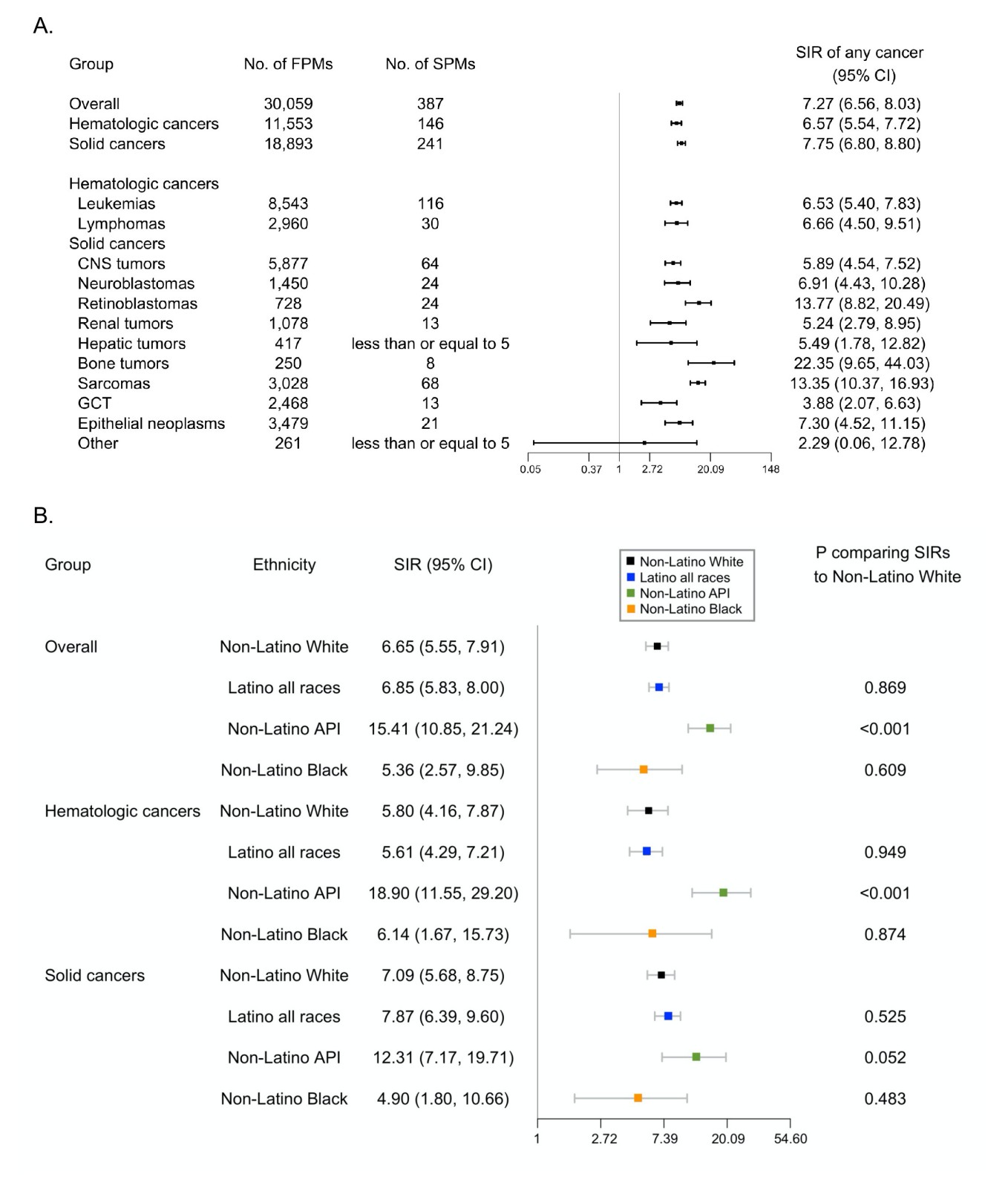

**Figure 3.** Relative risks of second primary malignancies. (**A**) Relative risks of second primary malignancies of any early-onset cancer (diagnosed under 26 years of age) given a proband with cancer, 1989–2015, California, USA. (**B**) Relative risks of second primary malignancies of any early-onset cancer (diagnosed under 26 years of age) given a proband with cancer by ethnic group, 1989–2015, California, USA. Cancers were classified into groups as defined by the International Classification of Childhood Cancer, Third edition (ICCC-3, November 2012) (https://seer.cancer.gov/iccc/iccc3.html).

*Figure 3 continued on next page*

*Figure 3 continued*

Hematologic cancers include leukemias and lymphomas. Solid cancers include CNS tumors, neuroblastomas, retinoblastomas, renal tumors, hepatic tumors, bone tumors, sarcomas, GCT, epithelial neoplasms, and others. The axis for SIR was natural log-transformed. SIR and 95% CI were not calculatable for cancers with zero observed case. p was calculated assuming a Poisson distribution. Abbreviations: SIR, standardized incidence ratio. CI, confidence interval. FPM, first primary malignancy. SPM, second primary malignancy. API, Asian/Pacific Islander.

lymphoma (*Chattopadhyay et al., 2018*), and neuroblastoma (*Applebaum et al., 2015*). The excessive familial risks of certain cancers are highly likely to be associated with genetic predisposition. The archetypic examples are germline loss-of-function mutations in *RB1*, which are found in ~40% of retinoblastoma cases (*Kamihara et al., 2017*), and adrenal cortical cancer, with germline *TP53* mutations accounting for most familial cases (*Kamihara et al., 2017*). Low penetrance common genetic variations, for instance in *CEBPE*, *IKZF1*, and *ARID5B* genes in ALL, are associated with cancer risk and may also contribute to familial concordance as combinations of low-frequency alleles or 'polygenic risk scores' have been shown to be as impactful as single strong predisposition mutations in adult cancers (*Fantus and Helfand, 2019*; *Yadav and Couch, 2019*); however, their contribution to cancer clustering among children and their families has not yet been studied.

Our data demonstrate a higher degree of familial-based clustering of solid cancers among Latinos and non-Latino Blacks compared to non-Latino Whites, and hematologic cancers among Asian/Pacific Islanders. This familial concordance is likely due to both shared genetic and environmental causes and appears to affect some cancer types differentially by race/ethnicity. Latinos are an admixed population, comprising an ancestral mixture from Native American, European, and African sources; likewise, non-Latino Blacks are admixed with African and smaller levels of European ancestry. Asian/Pacific Islanders constitute a particularly diverse group in California with origins in multiple countries. California Latinos, particularly the youth population, are largely from Mexico and harbor a higher risk of certain cancers particularly pediatric leukemias, the most common cancer in children (*Barrington-Trimis et al., 2017*); however, this higher risk is partially accounted for by a higher frequency of common risk alleles, which do not address strong familial predisposition loci (*Walsh et al., 2013*) and obviously cannot account for their higher familial risk of solid tumors. Clearly, this higher risk identified in relation to the family units in US minorities has not been systematically studied, and our results here beg for an analysis of comparative sources of genetic and environmental risk that contribute to the higher risk and familial clustering of certain cancers in Latinos, Blacks, and Asian/Pacific Islanders.

Therapy of the first primary cancer is a major factor in the induction of secondary independent malignancies (*McNerney et al., 2017*; *Mazonakis et al., 2018*; *Turcotte et al., 2019*). Multiple primary cancer diagnoses are considered a key feature of hereditary cancer predisposition syndromes (*Valdez et al., 2017*). As such secondary cancers are rare, genetics are still likely to play a strong role (*Churpek et al., 2016*), and our overall SPM results here emphasize a similar patterning as cancer clustering in FPMs. Of note, our analysis was not designed to distinguish risk influences from therapies for the primary cancers on secondary cancers.

For some tumor types, the germline predisposition was readily noted in this cohort, for example 10 of the 14 affected relatives who had a proband with retinoblastoma were diagnosed with the same cancer, an unsurprising finding given that germline *RB1* mutations account for a significant proportion of retinoblastoma are highly penetrant and those tumors tend to be diagnosed young. We also observed increased relative risks for sarcomas given a proband with leukemias, suggesting the presence of families with Li–Fraumeni syndrome, which is characterized by a spectrum of childhood and adult onset cancers including adrenocortical carcinoma, breast cancer, CNS tumors, sarcomas, and leukemia (*Valdez et al., 2017*).

Population-level selection pressures are thought to influence the relative frequencies of alleles. For instance, genetic adaptations that shaped the Native American genome to cold and warm environments (*Reynolds et al., 2019*), and immune response following colonization by Europeans (*Lindo et al., 2016*). Our result suggests that some adaptive selection pressures, or simply genetic drift, in specific ethnic groups may differentially influence familial cancer clustering as it does for immune and metabolic phenotypes among several ethnic groups (*O'Fallon and Fehren-Schmitz, 2011*; *Vatsiou et al., 2016*). If replicated in other study settings, this contrast between genetic risk

of child and adult onset cancers by ethnicity should be studied further for a fuller understanding of familial risks.

Our analyses capitalized on the highly diverse population in California, allowing us to quantify the relative risks across different ethnic groups. Moreover, the utilization of linked population-based registries in California enabled us to minimize the selection and information biases introduced by a case–control study design or other strategies that only sample portions of the population. There are also some limitations of our study. Despite the large number of total cancer cases, the number of affected siblings and second primaries is very small for some cancer types, thus limiting the power to detect significant relative risks. Also, we are unable to track cancer incidence for affected siblings, maternal cancers, and SPMs that may have been diagnosed outside of California. In addition, the follow-up time of 26 years is not enough for a comprehensive detection of SPMs in the probands, nor for cancers arising in proband mothers at older ages. These insufficient follow-up time and loss-to follow-up issues have limited our ability to quantify the relative risks among mothers with cancer onset at older ages (>40 years). Furthermore, it is likely that the low number of mothers with cancer is a result of bias against some very strong cancer predispositions, so the patients could not survive long enough or be healthy enough to reproduce. Lastly, the lack of records on fathers reduces our ability to quantify the relative risks among other first-degree relatives and may reduce the appreciation of the potential contribution of high-risk cancer predisposition syndromes which can be inherited from either parent.

Accepting those limitations with the current dataset, our study has several important implications that may open windows to future research. First, the genetic predispositions driving the excessive early-onset cancer risks among the Latino, non-Latino Black, and Asian populations, whether from higher frequencies of known cancer predisposition syndromes or mutations in novel genes or from a higher burden of common or rare genetic risk alleles, warrant further investigation. Second, the comparative attributable fraction of familial risk based on environmental risk factors interacting with genetic predispositions warrants further investigation. Lastly, descriptive studies on familial and secondary cancer risks among race/ethnic groups other than NLW may provide additional insights into cancer incidence variation leading to incidence disparities by race/ethnicity and provide critical information for tailoring appropriate messages for familial genetic counseling.

## Acknowledgements

Funding: This work was supported by the V Foundation for funding this work (Grant FP067172). The collection of cancer incidence data used in this study was supported by the California Department of Public Health as part of the statewide cancer reporting program mandated by California Health and Safety Code Section 103885; the National Cancer Institute's Surveillance, Epidemiology, and End Results Program under contract HHSN261201000140C awarded to the Cancer Prevention Institute of California, contract HHSN261201000035C awarded to the University of Southern California, and contract HHSN261201000034C awarded to the Public Health Institute; and the Centers for Disease Control and Prevention's National Program of Cancer Registries, under agreement U58DP003862-01 awarded to the California Department of Public Health. The ideas and opinions expressed herein are those of the author(s), and no endorsement by the California Department of Public Health, the National Cancer Institute, or the Centers for Disease Control and Prevention or their contractors and subcontractors is intended or should be inferred.

## Additional information

### Funding

| Funder | Grant reference number | Author |
| --- | --- | --- |
| V Foundation for Cancer Research | FP067172 | Andrew D Leavitt<br>Lucy A Godley<br>Joseph L Wiemels |

The funders had no role in study design, data collection and interpretation, or the decision to submit the work for publication.

## Author contributions
Qianxi Feng, Data curation, Software, Formal analysis, Investigation, Visualization, Methodology, Writing - original draft, Writing - review and editing; Eric Nickels, Data curation, Methodology, Writing - review and editing; Ivo S Muskens, Supervision, Visualization, Methodology, Writing - review and editing; Adam J de Smith, Conceptualization, Supervision, Writing - review and editing; W James Gauderman, Supervision, Methodology, Writing - review and editing; Amy C Yee, Resources, Project administration, Writing - review and editing; Charite Ricker, Writing - review and editing; Thomas Mack, Methodology; Andrew D Leavitt, Lucy A Godley, Conceptualization, Resources, Funding acquisition, Writing - review and editing; Joseph L Wiemels, Conceptualization, Resources, Supervision, Funding acquisition, Investigation, Methodology, Writing - review and editing

## Author ORCIDs
Qianxi Feng (ID) https://orcid.org/0000-0003-4888-4629
Joseph L Wiemels (ID) https://orcid.org/0000-0003-4838-9951

## Ethics
Human subjects: The collection of cancer incidence data used in this study was supported by the California Department of Public Health as part of the statewide cancer reporting program mandated by California Health and Safety Code Section 103885; the National Cancer Institute's Surveillance, Epidemiology, and End Results Program under contract HHSN261201000140C awarded to the Cancer Prevention Institute of California, contract HHSN261201000035C awarded to the University of Southern California, and contract HHSN261201000034C awarded to the Public Health Institute; and the Centers for Disease Control and Prevention's National Program of Cancer Registries, under agreement U58DP003862-01 awarded to the California Department of Public Health. The ideas and opinions expressed herein are those of the author(s), and no endorsement by the California Department of Public Health, the National Cancer Institute, or the Centers for Disease Control and Prevention or their contractors and subcontractors is intended or should be inferred.

## Decision letter and Author response
Decision letter https://doi.org/10.7554/eLife.64793.sa1
Author response https://doi.org/10.7554/eLife.64793.sa2

# Additional files

## Supplementary files
• Source code 1. Statistics and coding supplement.

• Supplementary file 1. Supplementary files. (A) Relative risks of the same type of early-onset cancer with the proband among siblings and mothers by ethnic group, 1989 to 2015, California, USA. (B) Relative risks of siblings and mothers for a specific type of early-onset cancer (diagnosed 0–26 years of age) given a proband with cancer, 1989–2015, California, USA. (C) Relative risks of any early-onset cancer (diagnosed at 0–26 years of age) for siblings and mothers of the same type of cancer with the proband given a proband with cancer by subgroups, 1989–2015, California, USA. (D) Relative risks of any early-onset cancer (diagnosed 0–26 years of age) among siblings and mothers by ethnic group, 1989–2015, California, USA. (E) Second primary malignancies in families exhibited familial risks and families did not exhibit familial risk. (F) Relative risks of second primary malignancies of the same type of early-onset cancer (diagnosed 0–26 years of age) with the first primary malignancy by ethnic groups, California, USA.

• Transparent reporting form

## Data availability
We have uploaded a blinded version of datasets used to generate the figures in this study to Dryad. https://doi.org/10.5061/dryad.80gb5mkq6.

The following dataset was generated:

| Author(s) | Year | Dataset title | Dataset URL | Database and Identifier |
|-----------|------|---------------|-------------|-------------------------|
| Feng Q | 2020 | Relative risks of familial cancers in California | https://doi.org/10.5061/dryad.80gb5mkq6 | Dryad Digital Repository, 10.5061/dryad.80gb5mkq6 |

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
