## [Decision Letter]

**Acceptance summary:**

This is a large population based study examining the familial risks of cancer in a California population under the age of 30. Risk of cancer is increased if a person has a first degree relative with cancer. Increased familial risk of cancer to first (and second) degree relatives is long established, but this is a robust study that furthers our understanding, specifically among Latinos, a common ethnic group in California.

**Decision letter after peer review:**

Thank you for submitting your article "Increased Burden of Familial-associated Early-onset Cancer Risk among Latino Americans Compared to non-Latino Whites" for consideration by *eLife*. Your article has been reviewed by 2 peer reviewers, and the evaluation has been overseen by a Reviewing Editor and Senior Editor. The following individual involved in review of your submission has agreed to reveal their identity: Steven Narod (Reviewer #1).

As is customary in *eLife*, the reviewers have discussed their critiques with one another. What follows below is my edited compilation of the essential and ancillary points provided by reviewers in their critiques and in their interaction post-review.

Essential revisions:

Please submit a revised version that addresses these concerns directly (see below). Although we expect that you will address these comments in your response letter, we also need to see the corresponding revision in the text of the manuscript. Some of the reviewers' comments may seem to be simple queries or challenges that do not prompt revisions to the text. Please keep in mind, however, that readers may have the same perspective as the reviewers. Therefore, it is essential that you attempt to amend or expand the text to clarify the narrative accordingly.

*Reviewer #2 (Recommendations for the authors):*

May major concern is with the methods used to compare the ethnic subgroups. This needs to be revisited.

It is distracting that the legend on the Forest plot is on the log-scale, but the numbers reported are on the original scale. Please change the numbers on the X axis to be on the original scale.

[Editors' note: further revisions were suggested prior to acceptance, as described below.]

Thank you for resubmitting your work entitled "Increased Burden of Familial-associated Early-onset Cancer Risk among Minority Americans Compared to non-Latino Whites" for further consideration by *eLife*. Your revised article has been reassessed by the a Senior Editor acting as Reviewing Editor.

The manuscript has been improved but there is a remaining issue that needs to be addressed, as outlined below:

Please revise the following statements in the abstract: "Higher relative risk of any cancer in siblings and mothers given a proband with solid cancer (P<0.05) was observed for both Latinos (SIR=4.98;95%CI:3.82-6.39) and for non-Latino Blacks (SIR=7.35;95%CI:3.36-13.95) compared to non-Latino White subjects (SIR=3.02;95%CI:2.12-4.16)." This statement needs to be rewritten as the single p value does not make any sense where it is placed.

---

## [Author Response]

Reviewer #2 (Recommendations for the authors):My major concern is with the methods used to compare the ethnic subgroups. This needs to be revisited.It is distracting that the legend on the Forest plot is on the log-scale, but the numbers reported are on the original scale. Please change the numbers on the X axis to be on the original scale.

We sincerely appreciate the comments from Reviewer 2. And we want to thank Reviewer 2 for pushing on the inconsistency between confidence intervals and p-value comparing the SIRs between race/ethic groups. While overlapping CI’s do not necessarily indicate a lack of significance in the effect sizes, the apparent contrast in these statistical measures was too extreme to be believable and indeed there was an error.

We reconstructed our data from scratch and recalculated all statistical comparisons with our statistician, Dr. W. J. Gauderman, and found a recurrent mistake in the calculation of p-value comparing the SIRs between race/ethic groups. We have corrected this mistake throughout the manuscript. Please refer to the new figure 1, 3, and supplementary materials for the corrected numbers. The p values are now somewhat attenuated, and significant differences between Latinos and NL whites persist for solid tumors. In addition, Asians have significantly increased familial risk for hematologic cancers, and non-Latino Blacks have significantly increased risk of solid tumors when compared to non-Latino whites. Because of this broader enhanced risk evident in minority groups (with the corrected statistical comparisons), the focus of the manuscript was changed slightly emphasizing higher risks among minority groups in respective hematologic and solid tumor categories. There were also SIR differences suggested between many individual types of cancer, while not reaching formal statistical significance.

[Editors' note: further revisions were suggested prior to acceptance, as described below.]

The manuscript has been improved but there is a remaining issue that needs to be addressed, as outlined below:Please revise the following statements in the abstract: "Higher relative risk of any cancer in siblings and mothers given a proband with solid cancer (P<0.05) was observed for both Latinos (SIR=4.98;95%CI:3.82-6.39) and for non-Latino Blacks (SIR=7.35;95%CI:3.36-13.95) compared to non-Latino White subjects (SIR=3.02;95%CI:2.12-4.16)." This statement needs to be rewritten as the single p value does not make any sense where it is placed.

Thank you for the issue brought up about the statement in the abstract. We agree that the placement of the p-value is unclear as to what it refers to. We therefore would like to change the sentence as follows:

"Both Latinos (SIR=4.98;95%CI:3.82-6.39) and non-Latino Blacks (SIR=7.35;95%CI:3.36-13.95) exhibited significantly higher relative risk of any cancer in siblings and mothers given a proband with solid cancer when compared to non-Latino White subjects (SIR=3.02;95%CI:2.12-4.16)."

In this sentence, we state that both Latinos and non-Latino Blacks have “significantly” higher risk, which assures the reader that the p value reaches a p=0.05 threshold. This significance refers to the fact that the Latino SIR (4.98) and non-Latino black SIR (7.35) were statistically significantly higher than white subject SIR (3.02). We feel that adding the actual p values to the sentence (within the quotes) will not make the point clearer, as readers may believe that P values would refer to the SIR being statistically different than the null (which they are, but that is not the point of the sentence).